# From Comparison to Confidence: The Dove Self-Esteem Project and the Transformation of Beauty Perceptions on Social Media

**DOI:** 10.3390/bs15040414

**Published:** 2025-03-24

**Authors:** Jihye Kim, Minseong Kim

**Affiliations:** 1Department of Integrated Strategic Communication, College of Communication and Information, University of Kentucky, Lexington, KY 40506, USA; jihye.kim@uky.edu; 2Department of Marketing and Information Systems, College of Business, Louisiana State University Shreveport, Shreveport, LA 71115, USA

**Keywords:** integrated marketing communication, brand authenticity, consumer engagement, social responsibility in branding, social media, sentiment analysis

## Abstract

This study examines The Dove Self-Esteem Project to evaluate its effectiveness as an integrated marketing communication (IMC) strategy and its impact on brand perception. Employing sentiment analysis, topic modeling, and word network analysis, we analyzed consumer comments on posts related to The Dove Self-Esteem Project on social media platforms to uncover sentiments and thematic patterns. The findings indicate a complex consumer reception characterized by a spectrum of emotional responses and discussions centered on authenticity, engagement, and social impact. This study highlights the nuanced role of authenticity in consumer-brand relationships and the importance of multifaceted engagement in brand strategies. Despite limitations such as sample representation and the constraints of sentiment analysis, this research provides valuable insights into the dynamics of socially responsible branding and its reception. Future research directions include longitudinal studies, cross-cultural analysis, and the exploration of behavior changes to deepen our understanding of the long-term effects of IMC campaigns on consumer perceptions and brand equity.

## 1. Introduction

The advent of social media has cast new light on the constructs of beauty and identity, particularly influencing the self-perception and self-esteem of adolescents and young adults ([14]; [38]). As adolescents and young adults navigate a virtual landscape replete with idealized representations of beauty, the resultant comparative and evaluative processes often engender dissatisfaction and anxiety regarding their own physical appearance ([1]; [9]). In reality, The Dove Self-Esteem Project’s latest findings underscore this societal shift, revealing that a significant portion of US teens—two in three girls—are engaging with social media for over an hour daily, eclipsing the time spent in face-to-face interactions with peers ([31]). This digital immersion subjects them to a relentless stream of beauty content, much of it promoting unattainable ideals, which is linked to the erosion of self-esteem in half the girls surveyed ([9]; [31]; [38]). The Dove brand’s research also highlights a path to empowerment: The majority of girls reported a positive change in self-image upon reducing their exposure to such content, with seven in ten feeling better after unfollowing idealized beauty accounts ([10]).

Our research seeks to shed light on the theoretical foundations that explain social media’s impact on the self-esteem and body image of young individuals, as well as to assess the tangible efficacy of Dove’s integrated marketing communication (IMC) strategy in altering and refining these perceptions ([1]; [14]). This digital dynamic positions The Dove Self-Esteem Project as a counter-narrative, one that challenges the prevailing beauty paradigms and advocates for a more inclusive and affirming approach to self-perception. Accordingly, the objective of this study is to deepen the theoretical conversation, providing a comprehensive evaluation of the psychological effects that Dove’s IMC strategies exert on consumers’ perceptions of beauty standards in the social media arena. In practice, our study critically evaluates Dove’s initiatives from the perspectives of corporate social responsibility (CSR) and branding, deriving recommendations grounded in evidence through an analysis of social media commentary on the project ([12]). Ultimately, our research aspires to promote a discerning and data-driven methodology in naturalistic settings to confront the extensive reach of these CSR-focused branding and IMC strategies on youth engaged with social media—settings that are more representative of real-world conditions than laboratory environments such as surveys or experimental designs. ([1]; [5]; [9]; [12]). By doing so, the current study aims to connect academic exploration with real-world industry practices, advancing the dialogue in the domain of self-esteem and body positivity promotion.

Specifically, The Dove Self-Esteem Project is a manifestation of the brand’s vision of beauty as a source of confidence rather than a cause for anxiety. By disseminating evidence-based tools and educational resources, Dove endeavors to mitigate the negative repercussions of digital media portrayals of beauty and to inculcate a resilient and positive body image among adolescents and young adults ([1]). The Dove Self-Esteem Project is committed to reaching a significant number of young individuals with self-esteem education via comprehensive IMC strategies, where the underlying mission transcends the mere promotion of products to encompass a broader social engagement ([10]). From a theoretical perspective, the introduction of Dove’s initiative presents an opportunity to explore the concept of brand authenticity within the context of social media ([37]). Brand authenticity is achieved when a brand’s communicative actions are congruent with its identity and values ([18]). As consumers increasingly seek out authentic brand experiences, especially within the domain of social media, Dove’s authentic portrayal of diverse beauty standards positions it as a brand that resonates with the public consciousness ([9]; [18]; [37]).

The methodological effort of our research contributes significantly to the academic discourse at the intersection of marketing communications, brand strategy, and consumer psychology. Its objective is to unravel the intricacies of socially responsible branding initiatives and their reception among digital consumer bases. This is pursued particularly within an observational framework, eschewing any theoretical manipulations typically employed in controlled environments, such as laboratory experiments ([14]; [38]; [43]). Through the meticulous analysis of social media commentary pertaining to The Dove Self-Esteem Project, conducted in the absence of researcher-imposed interventions ([43]), this study aims to provide a comprehensive appraisal of the campaign. It seeks to underscore the initiative as a quintessential model of IMC strategies, demonstrating profound brand engagement and contributing invaluable insights into the dynamics of consumer-brand interactions within the realm of digital platforms.

To address our research objectives, this study will dissect the consumer dialogue encircling The Dove Self-Esteem Project to unearth insights into its reception and resultant influence. Integrated marketing communication (IMC) emerges as a pivotal strategy for Dove’s Self-Esteem Project, blending traditional and modern media to create a unified and consistent message across all platforms. This paper extends the IMC discourse by examining its application on YouTube, a platform distinct in its interactive and content-rich nature. Unlike traditional media, YouTube’s dynamic environment, characterized by user-generated content and direct audience engagement, presents unique opportunities for IMC. By analyzing consumer comments on The Dove’s Self-Esteem Project, this study uncovers how IMC strategies leverage YouTube to foster deeper brand connections, thus filling a gap in the literature on IMC’s effectiveness in digital arenas. Specifically, utilizing sentiment analysis, topic modeling, and word network analysis to parse social media commentary, we aim to pinpoint the dominant emotions, recurring themes, and interaction trends that delineate the public’s interaction with Dove’s initiative on social media platforms. Our research marks a significant leap forward in the exploration of brand authenticity and consumer engagement on social media by deploying an advanced textual analysis approach that transcends traditional experimental designs and survey methodologies. By integrating sentiment analysis, topic modeling, and word network analysis, this study not only navigates the vast complexities of naturalistic data but also unveils nuanced insights into consumer perceptions, thereby advancing theoretical understandings in several key areas.

The sentiment analysis conducted in this study offers a nuanced measurement of consumer sentiment, capturing a spectrum of emotional responses. This methodological innovation allows for an in-depth exploration of the psychological mechanisms at play in consumers’ processing of brand-related information on social media. Theoretically, this contributes to the literature on brand authenticity by providing empirical evidence on how authentic brand narratives on social media can evoke a range of emotional responses, influencing consumer perceptions of brand authenticity.

Moreover, the employment of topic modeling techniques enables the identification of latent topics within consumer discourse, uncovering underlying themes that might not be apparent through surface-level analysis. This methodological advancement enriches the theoretical discourse on social media advertising effectiveness by revealing how consumers interact with and interpret the multifaceted nature of brand messages, offering insights into the types of content that resonate most with audiences. This adds a new dimension to our understanding of effective social media strategies that align with consumer expectations of brand authenticity.

Furthermore, the incorporation of word network analysis introduces a novel perspective on the structural aspects of consumer discourse. This approach illuminates the interconnectedness of themes, sentiments, and topics, offering a visual representation of the discourse ecosystem. From a theoretical standpoint, this analysis enhances our comprehension of the new psychological mechanisms in play as consumers process brand-related information, showcasing how the structure of social media discourse can influence consumer perceptions of brands. This insight contributes to the broader literature on consumer psychology by highlighting the role of networked narratives in shaping consumer engagement with brands on social media platforms.

Collectively, the methodological contributions of this study offer a comprehensive and advanced toolkit for dissecting the intricate fabric of social media discourse related to brand initiatives. Theoretically, these methodological advances facilitate a deeper understanding of brand authenticity, social media advertising effectiveness, and the psychological mechanisms underpinning consumer engagement. By bridging methodological innovation with theoretical advancement, our research not only enriches the academic discourse but also provides actionable insights for brands looking to navigate the complexities of consumer-brand interactions in the digital age. Through this integrated approach, this study makes a substantive contribution to the existing literature, advancing the dialogue in the domain of self-esteem and body positivity promotion and offering a foundation for future research in the field.

## 2. Literature Review

### 2.1. The Evolution of the Dove Self-Esteem Project

The Dove Self-Esteem Project, launched in 2004, is a long-standing corporate social responsibility (CSR) initiative aimed at redefining beauty standards by promoting body positivity, self-acceptance, and confidence-building among consumers ([10]). Unlike traditional beauty advertisements that emphasize idealized aesthetics, Dove’s campaign features real individuals from diverse backgrounds, aligning with broader brand activism and feminist advertising (femvertising) principles ([40]; [45]). Through digital and traditional media, Dove disseminates messages that challenge unrealistic beauty norms and promote authentic, unfiltered self-representation ([4]).

One of the most impactful campaigns, “Real Beauty Sketches”, exemplifies Dove’s approach by contrasting women’s self-perceptions with external perceptions from strangers, demonstrating a tendency for women to undervalue their appearance ([42]). The campaign’s viral success and extensive consumer engagement on social media underscore the public’s resonance with self-esteem advocacy ([4]; [42]). However, some scholars argue that while these campaigns foster positive engagement, they may also reinforce the commodification of self-esteem, as Dove ultimately remains a commercial brand with profit-driven objectives ([16]).

Dove’s IMC strategy reflects a shift toward digital media dominance, leveraging YouTube, Instagram, and Facebook to reach younger audiences while maintaining a presence in traditional media ([39]). Beyond advertising, Dove integrates educational resources, community workshops, and influencer collaborations to solidify its credibility as a brand committed to body positivity and mental well-being ([10]). This multi-platform strategy extends its brand message beyond commercial marketing, positioning Dove as a pioneer in socially responsible branding ([4]).

### 2.2. Corporate Social Responsibility (CSR), Femvertising, and Brand Activism

CSR has evolved from a supplementary corporate function to a core strategic pillar in contemporary brand management, with consumers increasingly evaluating brands based on their ethical commitments ([6]). Brand activism, which involves corporations taking public stances on social, environmental, or political issues, has become a key differentiation strategy ([25]). This is particularly relevant in femvertising, a marketing approach that integrates feminist themes to empower women and challenge gender stereotypes ([45]).

Dove’s Self-Esteem Project exemplifies the convergence of CSR, femvertising, and brand activism, positioning itself as a brand that champions inclusive beauty standards while actively engaging in self-esteem advocacy ([10]). However, some critics argue that femvertising can sometimes be performative, with brands capitalizing on feminist ideals while maintaining profit-oriented objectives ([40]). Additionally, the authenticity of such campaigns is questioned when parent companies, such as Unilever (Dove’s parent company), also market beauty brands that reinforce conventional beauty ideals ([25]). Despite these criticisms, empirical research supports the notion that consumers respond positively to perceived brand authenticity, reinforcing the strategic value of CSR-driven campaigns ([35]).

### 2.3. The Role of Body Positivity in Branding

The body positivity movement, rooted in the fat acceptance movement of the 1960s, has significantly influenced advertising strategies, leading to the emergence of body-positive branding ([42]). By rejecting narrow beauty ideals, brands leveraging body-positive messaging foster consumer trust and loyalty as audiences increasingly demand authentic representation in marketing ([4]; [21]).

Empirical research supports the psychological impact of body-positive branding. [3] ([3]) found that body-positive advertisements enhance consumer engagement, particularly when authenticity is perceived as high. Similarly, [8] ([8]) demonstrated that body-positive social media content improves body satisfaction and mood, particularly when visual and textual elements reinforce each other. However, body-positive advertising faces criticisms, including accusations of tokenism and commercializing self-acceptance as a branding tool ([3]). Some researchers caution that brands engaging in body positivity must demonstrate long-term commitments to inclusivity rather than employing one-off campaigns that risk being perceived as opportunistic ([8]).

### 2.4. The Importance of Brand Authenticity in Marketing Communication

Brand authenticity has emerged as a critical determinant of consumer trust, engagement, and long-term brand loyalty, particularly in an era where consumers demand greater transparency and ethical responsibility from corporations ([29]). Authenticity in marketing communication is defined as the alignment between a brand’s stated values, messaging, and actual business practices, ensuring that companies deliver on their promises and remain consistent across platforms ([16]). In CSR-driven campaigns—especially those that incorporate femvertising and brand activism—authenticity plays a pivotal role in determining whether consumers perceive the campaign as genuine advocacy or performative branding ([25]).

Recent studies highlight that authenticity in marketing is multidimensional. [16] ([16]) propose a six-dimensional framework for authentic femvertising, consisting of transparency, consistency, identification, diversity, respect, and stereotype-challenging content. According to their findings, consumers actively assess whether a brand truly embodies the feminist values it promotes or whether it is simply capitalizing on social movements for commercial gain—a phenomenon commonly referred to as “femwashing”. [29] ([29]) further demonstrate that perceived authenticity in femvertising significantly enhances brand attitudes and consumer advocacy, particularly among socially conscious consumers. However, these studies also caution that brands engaging in cause-driven marketing must ensure that their social advocacy extends beyond advertising campaigns into their corporate policies and business practices to maintain consumer trust.

For CSR-driven initiatives like Dove’s Self-Esteem Project, authenticity is both a strategic advantage and a challenge. The brand’s positioning as an advocate for body positivity and self-esteem requires consistent alignment between its messaging and corporate actions. [16] ([16]) argue that brands must demonstrate long-term commitments to social issues rather than engaging in one-off campaigns that may be perceived as opportunistic. This aligns with [25]’s ([25]) assertion that performative advocacy—where brands adopt social causes without meaningful action—can lead to consumer skepticism and brand distrust. In Dove’s case, while the Self-Esteem Project has received widespread consumer praise for its inclusive messaging and educational efforts, it has also faced criticism due to its association with Unilever, which markets other beauty brands that reinforce conventional beauty standards ([26]). This paradox highlights a key challenge in CSR branding—the need for corporations to ensure congruence between their advocacy campaigns and their broader brand portfolio to sustain credibility.

Moreover, authenticity in brand communication extends beyond messaging into how brands facilitate consumer participation. According to [30] ([30]), authentic engagement is strengthened when brands encourage consumer involvement, co-creation, and open dialogue, as opposed to merely presenting a pre-scripted corporate narrative. Dove’s approach, which integrates real consumer testimonials, interactive social media engagement, and partnerships with self-esteem educators, aligns with this principle, fostering greater consumer trust and emotional connection ([29]). However, the challenge remains in maintaining transparency and ensuring that the empowerment narrative is not diluted by commercial motives. As the discourse on femvertising and CSR-driven marketing evolves, scholars emphasize the importance of sustained corporate accountability, third-party verification of social impacts, and long-term investments in the causes that brands claim to support ([16]).

### 2.5. Integrated Marketing Communication (IMC) Strategies

IMC has evolved to accommodate the complexities of digital media ([15]). Contemporary IMC strategies emphasize a seamless, cross-platform approach, leveraging both traditional and digital media to ensure consistent messaging and brand engagement ([22]).

Dove’s Self-Esteem Project exemplifies an advanced IMC strategy, incorporating social media, influencer collaborations, educational programs, and video content to maximize consumer reach ([39]). YouTube, in particular, serves as a high-engagement platform, allowing for long-form storytelling and interactive brand engagement ([28]). Research suggests that social media-based IMC strategies enhance brand credibility and consumer sentiment, particularly when they facilitate active user participation ([11]). However, while Dove’s IMC approach has been successful in engaging consumers, it is important to recognize its reliance on digital media metrics. Future research should explore the long-term effectiveness of such campaigns beyond immediate social media engagement, assessing whether they lead to measurable changes in consumer attitudes and behaviors ([32]).

### 2.6. Theoretical Framework: Dove Self-Esteem Project and IMC Strategies

The Dove Self-Esteem Project extends beyond immediate brand marketing objectives, contributing to both societal impact and brand equity enhancement ([21]; [39]). By positioning itself as a socially responsible leader in the beauty industry, Dove leverages IMC strategies to promote self-esteem advocacy, challenge conventional beauty standards, and build consumer trust through ethical marketing practices ([1]). Unlike traditional advertising approaches that focus on product promotion, the Self-Esteem Project integrates educational content, influencer collaborations, and user-generated narratives across multiple digital platforms, particularly YouTube. This multi-channel, engagement-driven approach aligns with IMC’s fundamental principle of delivering a unified and strategically coordinated brand message across different communication channels ([15]). The IMC framework is particularly relevant in analyzing Dove’s campaign because it emphasizes message consistency, cross-platform engagement, and participatory consumer interaction, which are critical for building long-term brand trust and loyalty.

A key theoretical foundation supporting this study is [20]’s ([20]) brand resonance model, which explains how brands cultivate deep consumer relationships through strategic engagement and authenticity. Keller’s model outlines four progressive stages of brand-building: brand salience (awareness-building), brand meaning (associating the brand with core values), brand response (consumer emotions and attitudes toward the brand), and brand resonance (deep emotional connections and advocacy). Dove’s IMC strategy effectively progresses through these stages by employing interactive, participatory branding techniques that foster consumer involvement and long-term brand equity. For instance, campaigns such as “Real Beauty Sketches” and “Reverse Selfie” have successfully engaged consumers on social media through emotional storytelling and real-life narratives, reinforcing Dove’s positioning as an advocate for self-esteem ([42]). By integrating these IMC techniques, Dove has successfully strengthened consumer identification with its brand message, further demonstrating how brands that prioritize participatory engagement cultivate higher consumer loyalty and resonance ([20]).

Beyond brand resonance, the theory of authentic engagement ([30]) provides additional insights into how authenticity-driven IMC strategies enhance consumer trust and credibility. This theory argues that brands cultivate stronger consumer relationships when their messaging aligns with their corporate values, maintains transparency, and fosters active consumer participation. Authentic engagement is built on three key principles: message consistency across platforms, participatory consumer engagement, and emotional authenticity ([30]). In Dove’s case, its long-term commitment to self-esteem education, community-based initiatives, and diverse representation in advertising enhances its perceived authenticity ([16]). However, some scholars argue that femvertising campaigns must be cautious of perceived opportunism, as brands that incorporate feminist ideals without genuine corporate alignment risk being seen as engaging in “femwashing” ([40]). For Dove, its association with Unilever—a company that also markets brands reinforcing conventional beauty standards—presents an authenticity challenge ([25]). Addressing this, Dove has maintained transparent messaging and educational partnerships to reinforce the sincerity of its self-esteem advocacy ([21]).

Despite the growing body of literature on IMC strategies, digital branding, and CSR-driven marketing, there remains an academic gap in comprehensive consumer-focused research evaluating the effectiveness of IMC in fostering trust and engagement on social media. While prior studies have examined IMC’s role in brand performance metrics and sentiment analysis, fewer have explored how IMC strategies shape long-term consumer relationships through authenticity and participatory engagement ([1]; [39]). This study contributes to addressing this gap by applying IMC theory alongside Keller’s brand resonance model and the theory of authentic engagement, providing a deeper understanding of how digital branding campaigns strategically cultivate consumer trust, brand advocacy, and emotional loyalty. By grounding the analysis in these theoretical perspectives, this study offers a structured and critical interpretation of how IMC strategies influence consumer perceptions in CSR-driven marketing.

### 2.7. Research Questions Development

Central to our exploration is the concept of brand authenticity within social media contexts. As consumers increasingly value authentic brand experiences, Dove’s portrayal of diverse beauty standards exemplifies how authenticity can resonate with consumers and influence their perceptions of brand identity and values. This notion of authenticity aligns with the first research question, aiming to understand consumer perceptions of the Dove Self-Esteem Project’s authenticity and its alignment with the brand’s proclaimed values. Thus: we pose research question 1—“What specific elements of The Dove Self-Esteem Project’s social media communication are perceived by consumers as authentic, and how do these perceptions align with Dove’s stated values of real beauty and body positivity?”

Furthermore, the Dove Self-Esteem Project illustrates the effectiveness of Dove’s IMC strategy in delivering coordinated, measurable, and persuasive brand communication. The project’s measurable impact on brand awareness, consumer sentiment, and self-esteem education highlights the relevance of the second research question, focusing on the sentiments and thematic constructs prevalent within consumer commentary: therefore, we pose research question 2—“Using sentiment analysis, what are the predominant emotional responses and thematic constructs in consumer commentary on The Dove Self-Esteem Project across multiple social media platforms, and how do these sentiments correlate with specific campaign messages or features?”

Lastly, Dove’s initiative extends beyond immediate marketing objectives to contribute to broader societal conversations about beauty and self-esteem. This strategic application of IMC in fulfilling CSR objectives underscores the importance of the third research question, which seeks to assess the implications of consumer perceptions for Dove’s brand equity and the perceived success of its IMC strategies: we therefore pose research question 3—“How do consumer perceptions of authenticity and sentiment toward The Dove Self-Esteem Project impact measurable aspects of brand equity (e.g., brand awareness, loyalty, and perception) and what does this imply about the effectiveness of Dove’s IMC strategy in enhancing its corporate social responsibility profile?”

## 3. Methods

### 3.1. Data Collection

This study delves into the societal and marketing significance of employing IMC strategies on YouTube, using the Dove Self-Esteem Project as a case study. It highlights how IMC when executed on digital platforms like YouTube, transcends traditional advertising goals, influencing societal norms and individual behaviors. Our findings will suggest that by consistently portraying diverse and realistic beauty standards, Dove’s IMC strategy has contributed to a positive shift in societal beauty perceptions. This shift not only bolsters the brand’s image as a socially responsible entity but also sets a precedent for how brands can use IMC to contribute to societal well-being. Thus, this research underscores the dual significance of IMC in enhancing brand equity while championing social causes, providing valuable insights for marketers aiming to align brand messaging with societal values.

Our research embarked on a meticulous journey to collect primary data by delving into the realm of social media commentary, with a spotlight on YouTube. This platform was selected due to its prominence as a vibrant arena for brand communication and a nexus for consumer engagement. Our focal point was The Dove Self-Esteem Project, an array of initiatives crafted by Dove to foster self-esteem and advocate for body positivity. Dove’s advertising endeavors over the years have spanned various themes, including “Real Beauty”, “Courage is Beauty”, and “You are Beautiful”. To delineate The Dove Self-Esteem Project videos from other campaigns, we established a set of precise criteria.

The first step in our selection process was to identify videos with a direct connection to The Dove Self-Esteem Project. This included videos explicitly labeled as part of the initiative or containing clear mentions of the project within their content. Recognizing the core themes of self-esteem, body positivity, and the challenge against beauty stereotypes was paramount. Therefore, we also conducted a content analysis of the videos, selecting those that explicitly focused on these themes. Furthermore, we incorporated videos associated with Dove’s known Self-Esteem Project campaigns, as detailed on Dove’s official channels and campaign literature (https://www.dove.com/us/en/dove-self-esteem-project.html, accessed on 3 April 2024). However, our selection process also necessitated the establishment of exclusion criteria to refine our dataset further. Videos that primarily centered on product promotion without integrating the project’s key messages of self-esteem or body positivity were excluded. Similarly, videos that mentioned self-esteem or related concepts without a clear alignment to the specific objectives or messaging of The Dove Self-Esteem Project were omitted from our analysis. Additionally, in cases where multiple videos presented substantially similar content or messages, we evaluated them on a case-by-case basis, choosing only one representative video for inclusion to avoid redundancy in our data collection efforts.

As a result, we included YouTube videos, such as Toxic Influence: A Dove Film|Dove Self-Esteem Project (https://www.youtube.com/watch?v=sF3iRZtkyAQ, accessed on 1 May 2024) in our data analysis. In the documentary Toxic Influence, the dialogue between mothers and their adolescent daughters explores the pernicious beauty standards propagated on social media. This YouTube video critically examines dangerous trends such as “fitspiration”, “thinspiration”, and the encouragement of elective cosmetic procedures targeted at young females. It reveals mothers’ astonishment upon realizing the normalization of such detrimental beauty advice among their daughters, (1) prompting them to initiate crucial discussions regarding the content their daughters encounter on their social media feeds and (2) confronting societal beauty norms and prompting reflection on self-identification with beauty. Conversely, we excluded videos like direct product-focused ads that did not explicitly tie into the Self-Esteem Project’s core themes. Also, videos that contained messages of general positivity or empowerment but lacked a direct link to the project’s goals were not included in our dataset. This selective process ensured that our analysis remained focused and relevant to the objectives of our research.

The technical aspect of our data collection hinged on the use of Python 3.8.20 scripts, leveraging libraries such as requests for HTTP operations and BeautifulSoup for parsing the webpage’s HTML structure. The efficiency and systematic approach in fetching comments were further enhanced by employing the YouTube Data API, with a keen adherence to YouTube’s terms of service and the API’s rate limits. Following the collection phase, our dataset underwent a stringent cleaning process, which included the anonymization of personal information to uphold the privacy of individuals in alignment with ethical research guidelines and data protection laws. Text normalization techniques such as lowercasing, punctuation removal, and whitespace trimming were applied to prepare the data for subsequent analysis.

The initial crawling process yielded a significant volume of comments. Post the cleaning and anonymization process, 573 comments were deemed suitable for qualitative analysis. Since this study primarily focused on English-language comments, it is likely that a significant proportion of the comments originated from English-speaking countries, such as the United States, Canada, Australia, and other English-speaking regions. This sample size was deemed ample to yield meaningful insights into consumer perceptions and emotional responses toward The Dove Self-Esteem Project, balancing the depth of analysis with manageability for detailed qualitative exploration.

Utilizing Python tools designed for natural language processing (NLP), we embarked on a qualitative data analysis journey. This involved a meticulous examination of comments to extract prevalent topics, sentiment orientations, and emotional undertones, thus deepening our understanding of the public’s reception and interaction with The Dove Self-Esteem Project. The insights garnered from this analysis aimed to lay a robust foundation for the ensuing discussions and conclusions of our study, enriching the scholarly discourse surrounding brand-led self-esteem initiatives.

### 3.2. Data Analysis

This study employs sentiment analysis, topic modeling, and word network analysis to examine consumer engagement with Dove’s Self-Esteem Project on YouTube. These computational techniques allow for a large-scale, data-driven analysis of consumer sentiment and thematic content. While qualitative approaches such as content analysis or semi-structured interviews could provide deeper interpretative insights, they present several methodological limitations that make them less suitable for capturing the scope and nature of social media-based consumer engagement ([19]; [24]). First, interviews and manual content analysis are typically limited to small sample sizes, making them less effective for analyzing high-volume digital conversations. Second, content analysis relies on human interpretation, which can introduce inconsistencies in coding and may lead to variability in theme identification across different coders. Third, interviews provide in-depth personal perspectives, but they do not capture real-time public discourse, which is critical when analyzing dynamic, fast-moving digital brand campaigns. To mitigate these challenges while ensuring a rigorous and scalable analysis of consumer engagement, this study prioritizes computational methods as the most appropriate approach for evaluating IMC strategies on social media ([19]). However, we recognize that computational methods have limitations—they cannot capture the full depth of personal consumer experiences, nor do they allow for probing deeper into consumer motivations. To address this, we propose that future research complement computational analysis with qualitative interviews to balance large-scale analysis with deeper interpretative insights.

Our research conducted a descriptive analysis to identify the most frequently occurring words in consumer comments. Word frequency analysis offers a quantifiable measure of the most prominent terms within the dataset, serving as an initial step to identify key topics of discussion among consumers. It lays the groundwork by highlighting the most used words, which can signal the importance of certain themes or subjects within the consumer commentary ([46]). This analysis was carried out on both the preprocessed and non-preprocessed datasets to ensure a comprehensive understanding of the text data ([34]; [46]):(1)Word frequency count: This study calculated the frequency of each unique word in the dataset. This helped in identifying the most prominent terms and topics discussed by consumers.(2)Visualization: To aid in the interpretation of the results, this research generated visualizations, including a word cloud and a bar chart. The word cloud provided a quick and intuitive representation of word frequency, while the bar chart offered a clear and quantitative view of the top words.

In addition to the descriptive analysis, our study performed a thematic analysis to extract and summarize the main themes from the social media comments ([24]; [34]). The thematic analysis builds upon the insights gained from word frequency analysis by interpreting the context and nuances surrounding these frequently occurring words. It involves a qualitative review to categorize comments into themes, allowing for the exploration of patterns, topics, and sentiments expressed by consumers in a more detailed and nuanced manner ([34]). Together, word frequency and thematic analyses provide a solid foundation by identifying what is being discussed and understanding the broader context of those discussions ([24]):(1)Random sampling: The authors selected a random sample of comments to perform a qualitative review, which mitigated potential bias that could arise from handpicking comments.(2)Theme identification: By carefully reading through the sample comments, the authors identified recurring themes and patterns. These themes were then categorized based on the context and content of the comments.(3)Detailed exploration: For each identified theme, the authors conducted a detailed exploration to understand the underlying sentiments and opinions expressed by the consumers. This included examining the use of specific words and phrases that were closely related to each theme.

Sentiment analysis complements the findings from word frequency and thematic analyses by adding a layer of emotional understanding to the identified themes and subjects ([2]). By assigning a sentiment polarity score to each comment, this analysis enables researchers to gauge the overall sentiment (positive, negative, or neutral) of the consumer feedback related to specific topics ([19]; [24]). It enriches the understanding of consumer perceptions by revealing not just what topics are important but also how consumers feel about those topics, thus offering a more holistic view of consumer engagement with The Dove Self-Esteem Project. The sentiment analysis was conducted using the TextBlob library, which is a Python library for processing textual data. TextBlob offers a simple API to conduct common NLP tasks such as sentiment analysis, part-of-speech tagging, and more ([2]). The specific steps in the sentiment analysis were as follows ([19]; [24]; [46]):(1)Sentiment calculation: Each comment was passed through TextBlob’s sentiment analysis method, which assigned it a sentiment polarity score. This score ranges from −1 to 1, where −1 indicates a highly negative sentiment, 0 is neutral, and 1 is highly positive.(2)Score aggregation: The sentiment scores for all comments were aggregated to obtain an overview of the sentiment distribution across the dataset.(3)Visualization: A histogram was generated to visualize the distribution of sentiment scores, providing a graphical representation of the overall sentiment across the comments.

Prior to sentiment analysis, the following preprocessing steps were performed on the textual data to ensure consistency and accuracy ([19]; [46]): (1) normalization (i.e., the text was converted to a standard lowercase format to treat all variants of a word equally), and (2) cleaning (i.e., special characters, numbers, and punctuation were removed from the text because they are generally irrelevant to sentiment analysis).

While thematic analysis provides a manual, qualitative insight into the data, topic modeling offers a quantitative method to discover latent topics within the text ([17]). By using Non-negative Matrix Factorization (NMF) and TF-IDF vectorization, this approach identifies underlying themes that may not be immediately apparent ([33]). Topic modeling can uncover nuanced topics beyond the explicit categorizations of thematic analysis, thus providing a complementary perspective on the data ([33]). This method allows for the exploration of themes at a granular level, highlighting the diversity of consumer discourse and enriching our understanding of the scope of consumer concerns and interests ([17]):(1)TF-IDF vectorization: The text data was transformed into a term-document matrix using TF-IDF (Term Frequency-Inverse Document Frequency) vectorization, emphasizing words that were frequent in a comment but not across all comments.(2)Model fitting: The NMF algorithm was applied to the TF-IDF matrix, with the number of topics set to five. This number was chosen to balance the granularity of topics with the interpretability of results.(3)Word association: For each topic, the top 10 words with the highest association weights were identified. These words were deemed the most representative of their respective topics.(4)Analysis tools: The analysis was conducted using Python’s sklearn library, which provides efficient and straightforward methods for vectorization and NMF.

Word co-occurrence networks were constructed to visualize and analyze the relationships between words within the comments. Network analysis focuses on the relationships between words within the dataset, offering insights into how different terms and concepts are interconnected. This method is instrumental in visualizing the structure of discourse, illustrating how various themes, sentiments, and topics are linked. By analyzing word co-occurrence networks, researchers can identify clusters of related terms, revealing complex patterns of discourse that may not be evident through other analytical methods ([2]). Network analysis, therefore, adds a structural dimension to the textual analysis, enhancing the comprehension of the dynamics within consumer commentary ([2]). The following steps outline the methodology:(1)Co-occurrence determination: A sliding window approach was used to establish word co-occurrences. A window size of four words was initially chosen, where words within this range in the text were considered to be related.(2)Graph construction: An undirected graph was created where nodes represented words, and edges represented their co-occurrence within the comments.(3)Weight assignment: Edges were weighted based on the frequency of co-occurrence between nodes. Higher weights indicated a stronger or more frequent association.

Network Simplification

(1)Threshold filtering: Edges with weights below a certain threshold were removed to simplify the network and focus on stronger relationships.(2)Top words focus: The network was further simplified to include only the top words identified from the NMF topic modeling analysis.(3)Isolated node removal: Nodes not connected by edges were removed to enhance the clarity of the visualization.

*Visualization.* The network was visualized using a spring layout algorithm, which positions nodes in two-dimensional space based on their edge lengths and weights, aiming to reflect the structure of the network.

*Analysis Tools.* The analysis utilized Python libraries, including networkx for network construction and manipulation and matplotlib for visualization.

The integration of these four analytical methods ensures a comprehensive and multidimensional exploration of the textual data. Each method addresses different aspects of the data—ranging from the identification of prevalent terms and themes, the sentiment behind these discussions, and the uncovering of latent topics to the understanding of how these elements are interrelated. This holistic approach not only enriches the understanding of consumer perceptions and emotional responses towards The Dove Self-Esteem Project but also ensures a nuanced analysis that can support more informed and effective decision-making. By explicating how these methods complement each other, we delineate a clear path toward achieving a thorough understanding of the dataset, thereby justifying the relevance and value of employing multiple analytical approaches in our study.

## 4. Results

### 4.1. Word Frequency and Thematic Analysis

The comments on The Dove Self-Esteem Project show a strong focus on marketing, social media, and the campaign’s message. The word “Dove (*n* = 2088)” appears most frequently, indicating that many comments directly reference the brand. The terms “marketing (*n* = 1473)”, “social media (*n* = 782)”, “campaign (*n* = 699)”, and “project (*n* = 679)” suggest a discussion around Dove’s advertising and outreach efforts. Words such as “beauty (*n* = 626)” and “self-esteem (*n* = 695)” reflect the central themes of Dove’s campaign, which aims to promote positive self-perception and confidence (see Figure 1). The following are common themes in The Dove Self-Esteem Project:(1)*Marketing and engagement efforts* (top words: marketing, project, integrated, promotional, direct marketing, public relations, strategic): The theme showed a strong consumer recognition of Dove’s efforts to disseminate its campaign messages across various platforms. The comments often mentioned the brand’s utilization of multiple channels for communication. An example comment: “Dove does a great job using integrated marketing communications with this project using numerous promotional tools”.(2)*Perceived authenticity* (top words: authentic, genuine, real, well-being, cares, unscripted, real people): The comments frequently touched upon the genuineness and relatability of Dove’s campaign, mentioning the use of real-life scenarios and non-actors. Such a theme suggests that consumers value the brand’s approach to portraying everyday beauty and well-being concerns. An example comment is as follows: “It did not feel like Dove was just trying to sell their products…. It made the consumer feel like Dove cares for their actual well-being”.(3)*Ethical considerations and social impact* (top words: ethical, social responsibility, societal substance, positive self-image, impact): Discussions often extended beyond the campaign’s content to consider its broader societal implications. Many comments praised Dove for addressing issues of self-esteem and body positivity, suggesting an appreciation for the brand’s commitment to these values. This theme encapsulates the audience’s recognition of Dove’s ethical positioning and perceived contribution to societal well-being. An example comment is as follows: “The product itself isn’t the selling point, but the selling point is the purpose behind the product and what the brand stands for”.(4)*Educational content* (top words: guides, workshops, self-esteem, body confidence, educational, parents): The provision of resources such as guides and workshops was frequently acknowledged in comments, reflecting an appreciation for Dove’s efforts to support education on self-esteem and body image issues. This feedback suggests that the brand’s educational initiatives are a valued component of the campaign, resonating with the audience as a meaningful extension of its messaging. An example comment: “They also have many other sections that include confidence-building workshops”.(5)*Social media’s role* (top words: social media, influence, positive, educational, impact): The discussion around social media’s influence on beauty standards and self-perception was prominent. The comments often recognized Dove’s attempts to use social media as a positive force, highlighting the brand’s efforts to educate and inspire a healthier engagement with online content. This theme reflects consumer awareness of the challenges and opportunities presented by social media in shaping self-esteem. An example comment is as follows: “The campaign is relevant… [Dove] mentions how they have reached over 60 million young people and have the goal of reaching ¼ billion to have a positive body image”.(6)*Brand loyalty and positive impact* (top words: brand loyalty, positive effect, well-being, consumer relationship): While comments suggested a positive reception to Dove’s campaign, indicating an alignment with the brand’s values, it is essential to note that this theme reflects observed sentiments from social media interactions rather than a direct measurement of brand loyalty or consumer trust. An example comment is as follows: “It made the consumer feel like Dove cares for their actual well-being, and this can lead to a very positive effect on brand loyalty”.(7)*Comparison with other campaigns* (top words: comparison, other companies, product placement, Coca-Cola, Hello Happiness): Some comments drew comparisons between Dove and other brands or campaigns, often in a generalized manner. These observations suggest a perception of Dove’s unique approach to marketing and social advocacy. It is crucial to acknowledge that these comparisons are based on consumer perceptions, as expressed in the comments, without direct comparative analysis of Dove’s campaign against others. An example comment is as follows: “Most companies… always attempt to focus on their product regardless of the message they are attempting to convey”.

These themes portray The Dove Self-Esteem Project as a comprehensive, ethically driven campaign that leverages authenticity and social responsibility to connect with consumers on a deeper level than traditional advertising. The campaign’s use of real people, focus on educational resources, and understanding of social media’s role in shaping self-esteem are particularly noted for their positive impact on consumer perception of the Dove brand. Through these detailed thematic explorations, it becomes clear that The Dove Self-Esteem Project resonates with consumers on multiple levels, from its socially conscious messaging to its educational initiatives and its realistic approach to the role of social media in shaping self-image.

### 4.2. Sentiment Analysis

The sentiment analysis of the 583 consumer comments yielded the following results:(1)Count: A total of 583 comments were analyzed, with each comment receiving a sentiment score.(2)Mean sentiment score: The average sentiment score across all comments was approximately 0.159, which indicates a generally positive sentiment.(3)Standard deviation: The standard deviation of sentiment scores was 0.065, demonstrating moderate variability in sentiment across the comments.(4)Minimum sentiment score: The lowest recorded sentiment score was −0.035, denoting a slightly negative sentiment in that comment.(5)Maximum sentiment score: The highest sentiment score observed was 0.411, reflecting a strong positive sentiment.(6)Median sentiment score: The median score was 0.157, reinforcing the overall positive sentiment lean in the dataset.(7)Quartile distribution: The 25th percentile score was 0.119, and the 75th percentile score was 0.191, suggesting that the majority of comments fell into the positive sentiment category.

Figure 2 illustrates a right-skewed distribution, with the bulk of comments clustering on the positive side of the sentiment spectrum. These results confirm that the general consumer sentiment towards The Dove’s Self-Esteem Project is positive. The sentiment distribution indicates that the initiative is well-received among consumers, aligning with the positive themes identified earlier through this thematic analysis. This overall positive sentiment is indicative of Dove’s successful engagement with consumers on issues of social importance and self-esteem.

### 4.3. Topic Modeling

Topic modeling via NMF revealed five distinct topics across the consumer comments dataset. Each topic was characterized by a set of 10 words that were most frequently associated with that topic. The topics and their respective key terms are as follows:

Topic #01—Engagement strategy: This topic centers around discussions that recognize the variety of methods Dove employs to communicate its campaign messages. Key terms such as “customers”, “message”, “channels”, and “campaign” were prominent, indicating a focus on how Dove reaches its audience. This suggests that consumers notice and engage with the multi-platform strategy Dove uses, highlighting the breadth of the campaign’s reach.

Topic #02—Content and impact: Comments frequently mentioned “teens”, “beauty”, “social”, “educational”, and “inspiring”, pointing to discussions about the campaign’s content and its perceived influence on young audiences. This topic reflects consumer recognition of the campaign’s efforts to address beauty standards and the educational value of its messages, particularly in the context of social media.

Topic #03—Social media and guidance: Key terms like “media”, “social”, “parents”, “project”, and “girls” suggest that discussions often revolve around the influence of social media on youth and the role of parental guidance. This indicates an awareness among commenters of the complex dynamics between social media content and young people’s self-esteem, acknowledging Dove’s initiative as a positive counter-narrative.

Topic #04—Self-esteem and brand image: This topic, featuring terms such as “self”, “esteem”, “brand”, “positive”, and “image”, highlights conversations about the Dove brand’s association with promoting self-esteem and a positive self-image. The discussion suggests that the campaign is recognized for aligning Dove’s brand image with values of self-acceptance and body positivity.

Topic #05—Real beauty campaign: With terms like “beauty”, “women”, “real”, and “campaign”, this topic indicates discussions focused on Dove’s portrayal of “real beauty” and the response from consumers. It underscores the appreciation for Dove’s approach to challenging conventional beauty standards and promoting inclusivity.

The identified topics and their associated key terms provide a deeper understanding of what consumers discuss and prioritize in their comments about the Self-Esteem Project. They cover a range of aspects, from marketing strategies to the campaign’s impact on social perceptions and individual self-esteem. Figure 3 presents the top words for each of the five topics from the comments on The Dove Self-Esteem Project. Each subplot corresponds to one of the topics, showing the most significant words in descending order of weight.

The main purpose of this analysis was to provide a nuanced view of specific discussions surrounding self-esteem and body positivity. However, recognizing Dove’s long-standing commitment to challenging beauty norms through various campaigns, there might be the potential influence of Dove’s broader marketing narratives on the discussions identified. In other words, this study recognizes that Dove’s marketing narrative, especially the Campaign for Real Beauty, has cultivated a comprehensive brand image that could influence consumer perceptions and discussions, even in contexts specifically related to the Self-Esteem Project. For example, in Topic #5, we observed discussions that, while primarily centered on the Self-Esteem Project, occasionally mirrored sentiments and language characteristic of Dove’s broader campaigns, including the celebrated Campaign for Real Beauty. Terms such as ‘real beauty’ and discussions around body positivity in the context of the Self-Esteem Project occasionally reflected broader narratives that have been prominent in Dove’s advertising efforts over the years. This overlap suggests a cohesive brand image that Dove has successfully cultivated, where initiatives like the Self-Esteem Project are not seen in isolation but as part of a broader, consistent commitment to challenging beauty norms. It underscores the strength of Dove’s branding strategy, where different campaigns contribute cumulatively to the brand’s social mission, thereby enhancing consumer engagement and brand loyalty.

### 4.4. Network Analysis

To analyze the word co-occurrence network in consumer comments, this study utilized Python’s networkx library for constructing and visualizing the relationships between key terms. A sliding window approach with a window size of four words was applied to determine co-occurring word pairs, ensuring that terms frequently mentioned together were captured within a structured network. The nodes in the network represent the most commonly used words in the dataset, while the edges indicate their co-occurrence relationships. To enhance clarity, edges with lower weight thresholds were filtered out, allowing the network to highlight the strongest associations among key terms. The visualization was further refined using a spring layout algorithm, which optimally positions nodes based on their connectivity patterns, making the structure of consumer discourse more interpretable. The entire network visualization process, including data preprocessing and a graphical representation, was implemented using Python’s networkx and matplotlib libraries, ensuring a systematic and replicable methodological approach for identifying dominant themes in the dataset. The resulting network consisted of nodes representing the top words from each topic identified in the previous topic modeling analysis and edges representing significant co-occurrences. These are the network characteristics:(1)Degree of connectivity: Nodes displayed varying degrees of connectivity, with some nodes acting as central hubs within the network due to their higher degree of associations.(2)Network simplification: The application of a weight threshold and the removal of isolated nodes resulted in a clearer network, emphasizing the most significant word associations.

The network visualization demonstrated that certain words served as pivotal points of discussion among consumers, reflecting the central themes in the dataset. Words with higher degrees of connectivity were often related to core aspects of The Dove Self-Esteem Project, such as “beauty”, “self”, “campaign”, and “social media”, indicating the prominence of these themes in the consumer discourse (see Figure 4).

## 5. Conclusions and Implications

### 5.1. Analytical and Theoretical Framework for Understanding IMC Strategies in Digital Branding

While our findings provide a descriptive overview of consumer engagement with Dove’s Self-Esteem Project, it is crucial to situate these findings within a broader analytical and theoretical framework to fully understand the strategic significance of Dove’s IMC approach. Existing literature on IMC strategies in digital branding suggests that effective consumer engagement extends beyond mere message dissemination, requiring brands to establish emotional connections, perceived authenticity, and participatory co-creation to drive long-term brand loyalty ([15]; [20]). [20]’s ([20]) brand resonance model provides a useful framework for analyzing how Dove’s IMC strategy fosters consumer loyalty and advocacy. The model outlines the four following progressive stages: brand salience (awareness-building), brand meaning (developing brand associations), brand response (emotional and cognitive reactions), and brand resonance (deep consumer-brand connections and advocacy). Our findings suggest that Dove’s use of real consumer stories, interactive campaigns, and digital storytelling effectively builds brand resonance, as evidenced by the high level of positive consumer sentiment and active participation in discussions on self-esteem and beauty representation. By leveraging multi-platform engagement strategies, Dove successfully guides consumers through the brand resonance process, fostering a sense of community and shared values around body positivity.

Beyond brand resonance, the theory of authentic engagement ([30]) provides additional insight into how authenticity-driven IMC strategies enhance consumer trust and credibility. According to this framework, brands that demonstrate consistency between their CSR messaging, corporate values, and business practices cultivate stronger consumer relationships and deeper trust. Our findings indicate that Dove’s commitment to body positivity, educational initiatives, and diverse representation enhances its perceived authenticity, aligning with the theory’s emphasis on transparent and value-driven brand engagement. However, some scholars caution that femvertising campaigns, despite their empowering narratives, may still reinforce commercialized beauty norms ([40]), raising concerns about whether Dove’s self-esteem advocacy is genuinely altruistic or strategically profit-driven ([25]). This underscores the delicate balance CSR-driven brands must maintain—while Dove successfully engages consumers through participatory marketing and positive messaging, maintaining authenticity and credibility remains an ongoing challenge in the broader landscape of corporate brand activism.

By integrating Keller’s brand resonance model and the theory of authentic engagement, this study moves beyond descriptive findings to offer a structured interpretation of Dove’s IMC success. Our findings demonstrate that brand resonance is achieved through cross-platform engagement, emotionally compelling storytelling, and user-generated content, all of which contribute to deeper consumer trust and brand loyalty. Additionally, the integration of participatory marketing techniques and transparent messaging strengthens consumer perceptions of authenticity, reinforcing the strategic value of authenticity-driven IMC strategies ([16]; [29]). These insights highlight the importance of IMC as a holistic approach to consumer engagement, where success is not merely measured by message reach but by the depth of brand-consumer relationships cultivated through strategic digital branding efforts.

### 5.2. Theoretical Implications

Our empirical findings indicate that consumers perceive Dove’s Self-Esteem Project as an authentic initiative due to its genuine portrayal of diverse beauty standards and alignment with the brand’s core values. This perception is evidenced by the sentiment analysis results, which revealed predominantly positive consumer sentiment, particularly toward the campaign’s message of inclusivity and empowerment. Furthermore, the thematic analysis identified that key themes such as brand authenticity, ethical marketing, and self-esteem education were frequently discussed in consumer comments. These empirical results align with [18] ([18]) and [30] ([30]), who assert that brand authenticity is essential in fostering consumer engagement and trust. Moreover, the positive sentiment toward Dove’s messaging, as observed in our dataset, supports [35]’s ([35]) argument that authenticity in corporate social responsibility (CSR) enhances brand loyalty.

However, while these findings provide empirical support for the role of authenticity in consumer engagement, they do not directly establish causality between Dove’s campaign and long-term consumer behavior. Although the sentiment analysis suggests that consumers appreciate Dove’s messaging, it does not confirm whether this sentiment translates into increased brand loyalty, purchase intent, or lasting self-esteem improvements. Future studies employing longitudinal designs or experimental methodologies would be needed to assess whether exposure to such campaigns leads to measurable changes in consumer behavior over time. Thus, while this study contributes to the theoretical discourse on brand authenticity within socially responsible campaigns, its findings should be interpreted within the scope of platform-specific consumer engagement rather than definitive brand impact ([18]; [30]).

Second, our empirical analysis of consumer comments suggests that Dove’s campaign leverages social media effectively to promote self-esteem and body positivity, countering the negative effects of idealized beauty standards prevalent online. Specifically, sentiment and topic modeling analyses indicate that consumers recognize Dove’s efforts to provide a more inclusive and positive representation of beauty. This supports [13]’s ([13]) theory of social comparison, as Dove’s campaign challenges traditional comparison mechanisms by presenting alternative narratives of beauty. Moreover, the observed engagement with Dove’s campaign themes aligns with [44]’s ([44]) findings on how positive social media content can mitigate harmful comparison behaviors.

However, while our empirical results show that consumers engage with Dove’s campaign positively, they do not confirm whether the campaign has a sustained impact on self-esteem or body image over time. Social media engagement does not inherently translate to deeper psychological or behavioral changes ([27]). Future research could explore experimental or longitudinal methodologies to measure whether exposure to such campaigns leads to lasting improvements in self-perception and well-being. Thus, while this study enriches the theoretical discussion on social media’s role in shaping consumer psychology, its findings should not be overgeneralized beyond the context of digital discourse ([23]; [36]; [44]).

Third, our findings indicate that Dove’s educational resources serve as a critical component of its IMC strategy, with many consumers positively acknowledging the brand’s commitment to self-esteem education. Thematic analysis revealed that discussions frequently centered around Dove’s provision of workshops, guides, and online educational content, highlighting its perceived impact on raising awareness of body image issues. These empirical findings align with [7] ([7]) and [41] ([41]), who suggest that educational marketing strategies can shape consumer attitudes and behaviors.

Nevertheless, while consumers appear to appreciate Dove’s educational approach, this study does not measure the actual effectiveness of these educational efforts in changing consumer knowledge, attitudes, or behaviors. Future research should incorporate controlled studies or knowledge assessments to determine whether such educational resources lead to measurable improvements in self-esteem or body image awareness. Therefore, while our theoretical contribution highlights the role of educational content in IMC strategies, further empirical validation is needed to confirm its practical effectiveness in driving long-term social change ([7]; [41]).

Fourth, our sentiment analysis results indicate a generally positive emotional response toward the Dove Self-Esteem Project, suggesting that consumers engage with CSR-driven marketing favorably. However, it is crucial to recognize that sentiment analysis is a surface-level measure of consumer response and does not fully capture the depth of consumer sentiment, potential skepticism, or underlying motivations ([19]; [24]; [46]). While the positive sentiment distribution suggests that Dove’s campaign resonates with consumers, our study does not assess whether this engagement translates into behavioral loyalty, advocacy, or purchasing decisions.

Furthermore, sentiment analysis does not account for potential biases in social media discourse, as consumers who hold critical views of Dove’s campaign may be underrepresented in the dataset. Prior research suggests that CSR campaigns often face scrutiny from skeptical consumers who may perceive them as opportunistic rather than authentic ([16]). Future research should incorporate mixed-method approaches such as in-depth consumer interviews or survey-based studies to better understand how consumers interpret and react to CSR-driven branding initiatives over time.

Lastly, while the findings of this study indicate a generally positive consumer reception toward Dove’s Self-Esteem Project, it is essential to acknowledge alternative perspectives and potential skepticism surrounding the campaign. CSR initiatives, particularly those embedded in marketing campaigns, often face scrutiny regarding their authenticity and underlying motives. Some consumers may view Dove’s advocacy for body positivity as a strategic branding effort rather than a purely altruistic endeavor, raising concerns about “femwashing” or the commercialization of social issues ([16]). Prior research on CSR-driven marketing suggests that consumers are more likely to trust a brand’s social initiatives when they perceive consistency between the brand’s messaging and its corporate practices ([25]). However, Dove’s parent company, Unilever, owns brands that have historically promoted traditional beauty standards, leading some consumers to question the sincerity of Dove’s body-positive messaging. Additionally, some scholars argue that femvertising campaigns, despite their empowering narratives, may still reinforce beauty standards by centering appearance as a core aspect of women’s value ([40]). While our sentiment analysis indicates an overall positive consumer perception, it does not capture the voices of skeptical consumers who may express their critiques outside of brand-affiliated platforms. Future research could benefit from various methods to explore consumer skepticism in greater depth and assess how such perceptions influence brand trust, purchase intention, and long-term engagement.

### 5.3. Managerial Implications

From a practical perspective, the empirical findings highlight how Dove’s genuine portrayal of diverse beauty standards resonates with consumers, showcasing the power of authenticity in brand communication. This finding suggests that beauty brands should look beyond traditional marketing narratives and instead focus on real and relatable stories that reflect their brand values. Brands can achieve this by launching customer-driven storytelling campaigns, where real consumers share personal experiences of overcoming beauty stereotypes. To ensure credibility, brands should invite consumers to submit unedited video testimonials, which can be featured in campaign materials across digital and traditional platforms. Companies can also create long-form documentary content showcasing real people’s journeys toward self-confidence, reinforcing a brand’s commitment to authenticity beyond advertising.

Second, the effectiveness of Dove’s IMC strategy, particularly its use of digital platforms like YouTube, underscores the importance of a multi-channel, integrated approach to brand messaging. Beauty brands can take this insight further by creating campaign ecosystems that blend owned, earned, and paid media to drive engagement. Brands should design IMC campaigns that integrate video storytelling on YouTube, social media-driven user engagement, and collaborations with influencers or advocates in body positivity movements. For example, launching a “Beauty Without Filters” challenge on TikTok and Instagram—where users are encouraged to post unedited photos/videos—could serve as a powerful interactive strategy that generates organic engagement. Paid media, such as targeted advertising, should focus on boosting visibility for user-generated content rather than polished brand messages, fostering authentic audience participation.

Third, our research underscores the critical role of social media in shaping consumer perceptions of beauty and self-esteem. To capitalize on this, beauty brands should focus on community-driven engagement rather than one-way messaging. Companies should establish dedicated online communities (e.g., private Facebook groups, Discord channels, or brand-hosted forums) where consumers can discuss self-esteem, body positivity, and mental well-being. This could be further enriched by hosting monthly live Q&A sessions featuring psychologists, educators, or self-esteem advocates who address beauty-related concerns in a way that fosters trust and loyalty. Additionally, brands can partner with non-profits specializing in adolescent mental health and self-esteem education, integrating these collaborations into long-term IMC strategies rather than one-off campaigns.

Fourth, the sentiment analysis findings in this study indicate a generally positive reaction toward Dove’s campaign, with key themes focusing on authenticity, brand ethics, and positive messaging. This suggests an opportunity for brands to continuously refine their IMC strategies based on real-time sentiment tracking. Beauty brands should implement AI-powered consumer sentiment monitoring tools (e.g., Brandwatch, Sprinklr, or Socialbakers) to track audience reactions in real time. By analyzing consumer sentiment across different digital platforms, brands can adjust messaging, fine-tune content strategies, and identify potential areas of criticism before they escalate into public skepticism. A rapid-response strategy should also be in place, ensuring that brands can engage with consumers directly if concerns about authenticity or purpose-driven marketing arise.

Fifth, our thematic analysis highlighted the growing consumer demand for brands to contribute to societal well-being in a genuine and measurable manner. Brands cannot afford to engage in “performative activism” without long-term commitments. Beauty brands should institutionalize their social advocacy by embedding permanent self-esteem or diversity programs into their core business strategy. For instance, brands can introduce free digital self-esteem workshops through YouTube or develop educational partnerships with schools to provide body positivity curricula. Moreover, brands should commit to third-party audits of their CSR initiatives—allowing independent watchdog organizations to verify whether their actions align with their messaging. Publishing annual impact reports that document the number of individuals reached, funding allocated to advocacy initiatives, and measurable outcomes would significantly enhance credibility and transparency.

Lastly, our findings suggest that product and marketing innovations should align with evolving consumer expectations for authenticity, ethical practices, and social responsibility. Brands should develop inclusive product lines that cater to diverse beauty needs, emphasizing the use of unretouched product visuals in their advertisements. Additionally, sustainability should be a core part of product innovation—offering eco-friendly packaging, cruelty-free formulations, and carbon-neutral supply chains. Furthermore, brands can introduce interactive shopping experiences where customers can engage with AI-powered beauty personalization tools that suggest products based on skin type, tone, and real-life beauty concerns rather than unrealistic beauty standards. By integrating these ethical and consumer-driven elements into their marketing and product development, brands can reinforce a long-term authentic and purpose-driven identity.

### 5.4. Limitations and Directions for Future Research

Our study of The Dove Self-Esteem Project, through an examination of consumer comments, comes with inherent limitations that are important to acknowledge for a comprehensive understanding. The sample of comments may not accurately represent the broader population’s views because they are self-selected and potentially biased toward more vocal individuals. The scope of data, confined to the dataset at hand, might have missed pivotal insights from other platforms or offline discussions. Automated sentiment analysis tools, while useful, are not infallible in capturing the subtleties of human emotion, which can lead to misinterpretations. Given the temporal nature of the data, it is also possible that the current study does not reflect evolving consumer attitudes that shift with cultural trends. In addition, the diverse cultural and demographic perspectives that influence perceptions of beauty and self-esteem may not be adequately represented and controlled in the dataset. While quantitative methods were heavily employed, qualitative insights, which are crucial for understanding the complexities of consumer sentiment, were not the primary focus.

To move forward, the approaches of future research should aim to address the aforementioned limitations. First, longitudinal studies would offer valuable insights into how consumer perceptions of The Dove Self-Esteem Project have evolved over time. A cross-cultural analysis could provide a more global view of the campaign’s reception and relevance. By broadening the scope of data collection to encompass diverse platforms and offline channels, future studies could gain a more rounded understanding of consumer perceptions. Second, in-depth qualitative research methods such as interviews or focus groups would offer richer insights into consumer motivations and responses to the campaign. Experimental designs could shed light on the causal effects of the Self-Esteem Project on attitudes and behaviors among social media users. Focusing on specific demographics could reveal nuanced views and impacts across different segments of the population. Assessing whether positive sentiment translates into consumer behavior, such as purchase or advocacy, is another critical avenue for research. Third, integrating this sentiment analysis with sales and performance data could help correlate campaign effectiveness with Dove’s financial success. An exploration of negative sentiments would also be beneficial, providing Dove with the opportunity to refine their campaigns. Lastly, understanding how Dove’s project influences competitor strategies and consumer perceptions could offer a broader perspective on the campaign’s industry-wide impact.

## Figures and Tables

**Figure 1 behavsci-15-00414-f001:**
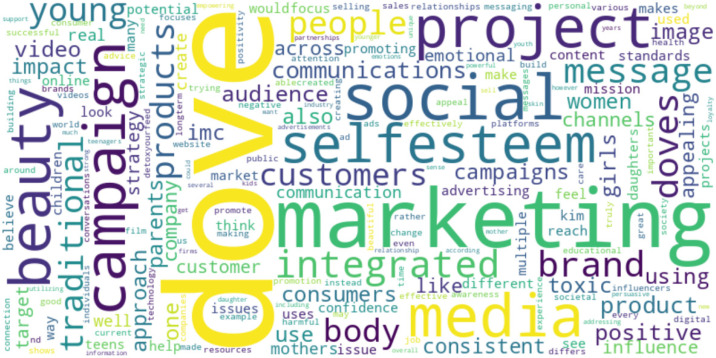
Words clouds with the most common terms from the comments appearing larger.

**Figure 2 behavsci-15-00414-f002:**
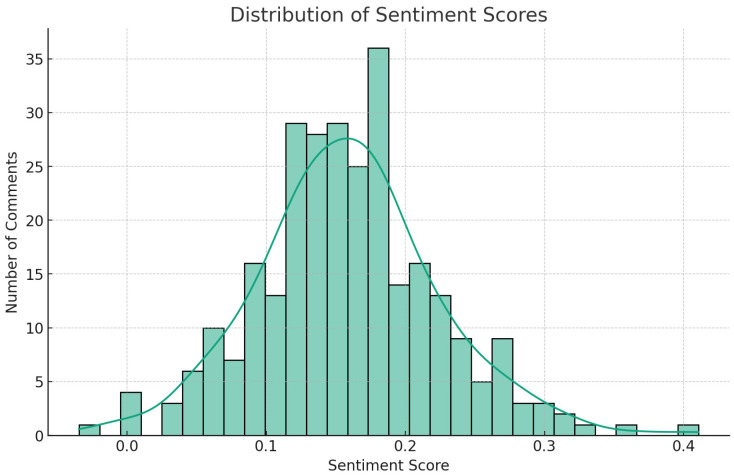
The result of sentiment analysis.

**Figure 3 behavsci-15-00414-f003:**

Topics in Dove’s Self-Esteem Project comments.

**Figure 4 behavsci-15-00414-f004:**
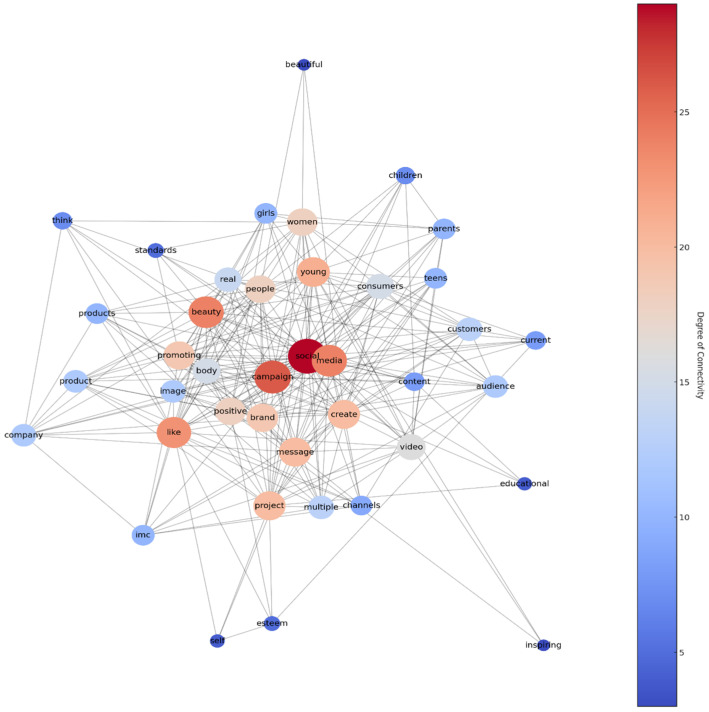
Word co-occurrence network with degree highlighting.

## Data Availability

The data presented in this study are available on request from the corresponding author.

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
