# Peer review of "From Comparison to Confidence: The Dove Self-Esteem Project and the Transformation of Beauty Perceptions on Social Media"

_behavsci, 2025, doi:10.3390/bs15040414_

Round 1

Reviewer 1 Report

Comments and Suggestions for Authors

This is an interesting study focused on using IMC strategies on YouTube with the Dove Self-Esteem project used as a case study. The findings and the study itself are very descriptive and it would be better if a more analytical and critical framework could be adopted to situate the study and its importance. I can see that different types of analyses were done but they all seem very descriptive.

This study explores the effectiveness of The Dove Self-Esteem project using the Integrated Marketing Communication (IMC) strategy. While it is understandable that this is a case study, a strong justification needs to be given as to why other strategies or models were not used. This study needs to have a strong theoretical underpinning which I do not see in its current form.  

It would have been better to employ content analysis to analyse consumer comments on posts on YouTube. Semi-structured interviews could have been used to explore participants’ perceptions of The Dove Self-Esteem project. The use of sentiment analysis, topic modelling and word network analysis offers very descriptive findings, making it difficult to draw valid conclusions from the study.   

Authenticity in consumer-brand relationships is indicated as a key finding but prior literature on this needs to be explained in detail. Some information on this is given on pages 5-6 but this is not sufficient. A more critical review of the literature is required.  

The study in its current form is very descriptive without a strong theoretical underpinning while the conclusions do not add something new to the existing literature. The authors should frame the study better and employ more rigorous methods to offer in-depth insights into the findings.

Reviewer 2 Report

Comments and Suggestions for Authors

Dear authors, 

First of all, thank you for the opportunity to evaluate such an interesting paper. Considering the paper and its potential impact, I recommend the following minor changes: 

  1.  The literature review section should be divided into subsections / topics - the current setting is hard to follow
  2. In section 3.1 - could you also mention, if possible, the geographical spread of the comments
  3. In the 4.1 word frequency ....., you mentioned multiple words as being the most frequent, but you did not provide any numerical or percentage occurrences
  4. For the data of figure 4, could you please also mention the tool you have used to develop the network word co-occurrence

Looking forward to receiving the improved version! 

Reviewer 3 Report

Comments and Suggestions for Authors

The study is well-structured and contributes meaningfully to the academic discourse on brand authenticity, social responsibility, and digital marketing.

That said, there are areas where the manuscript could be further refined.

The literature review is comprehensive and well-supported by relevant citations. However, some sections could benefit from more synthesis and a clearer connection between theoretical foundations and the study’s objectives. For example, while the discussion of body positivity and femvertising is important, certain paragraphs (e.g., page 5, lines 150–170) repeat key arguments without progressing the discussion. Consider condensing overlapping ideas and ensuring each section directly contributes to the study's core research questions.

The discussion and interpretation of results are insightful but could be more balanced by acknowledging alternative perspectives and potential limitations. While the study successfully demonstrates positive consumer reception, it does not sufficiently explore potential skepticism or criticisms of the campaign. Some consumers might perceive Dove’s strategy as commercializing social issues, a common critique of corporate social responsibility (CSR) initiatives. Addressing these nuances would strengthen the academic rigor of the discussion.

The study’s practical contributions are valuable, but certain areas could be further developed to provide actionable insights for marketers and policymakers. For example, the study discusses how consumer perceptions of authenticity impact brand engagement, but it could go further in providing concrete recommendations for brands looking to implement similar IMC strategies.

The conclusion effectively summarizes the study but could be more explicit in distinguishing between empirical findings and theoretical inferences. While the results indicate a positive sentiment toward Dove’s campaign, the manuscript occasionally makes broad generalizations about its overall effectiveness without fully addressing potential limitations.

I look forward to seeing this research contribute to important discussions on brand authenticity, digital marketing, and social impact. Keep up the great work, and best wishes for the revision process!
